# Effects of Cortactin Expression on Prognosis in Patients with Breast Cancer

**DOI:** 10.3390/diagnostics13182876

**Published:** 2023-09-07

**Authors:** Hwangkyu Son, Seungyun Jee, Hyebin Cha, Kihyuk Song, Seongsik Bang, Hyunsung Kim, Seungsam Paik, Hosub Park, Jaekyung Myung

**Affiliations:** Department of Pathology, Hanyang University College of Medicine, Seoul 04763, Republic of Korea

**Keywords:** cortactin, breast cancer, triple-negative breast cancer

## Abstract

Background: Cortactin is overexpressed in several types of invasive cancers. However, the role of cortactin expression in breast cancer prognosis has not been sufficiently elucidated. Therefore, we investigated the clinicopathological significance of cortactin in breast cancer. Methods: Tissue microarrays were prepared from a cohort of 506 patients with breast cancer, and cortactin expression was evaluated using immunohistochemistry. The cortactin immunoreactivity score (IRS) was quantified as the product of the intensity score and the percentage of immunoreactive cells. Cortactin expression was classified as low or high using the IRS (IRS ≤ 4 as a cortactin-low value and IRS > 4 as a cortactin-high value). We compared cortactin expression and clinicopathological factors according to the molecular subtypes of breast cancer. Results: Of 506 breast cancer cases, 333 and 173 showed high and low cortactin expression, respectively. Of the 333 patients with high cortactin expression, 204, 58, and 71 had luminal, HER2, and triple-negative breast cancer (TNBC), respectively. In the univariate and multivariate analyses of patients with TNBC, cortactin expression was found to be a significant prognostic factor for overall survival (OS). However, in all patients with non-TNBC, cortactin expression had no significant association with prognosis or overall survival. Survival curves revealed that among patients with TNBC, the high-cortactin group had a better prognosis in disease-free survival and OS. Conclusions: Cortactin expression may be a good biomarker for predicting the prognosis of patients with TNBC.

## 1. Introduction

Breast cancer is one of the most common types of cancer in women worldwide [1]. As per the 2020 global cancer statistics, 2.6 million (11.7% of all new cancer cases) women were diagnosed with breast cancer, and 680,000 women died of breast cancer [1]. Despite the advances in treatment outcomes based on an increased understanding of breast cancer pathogenesis, therapeutic limitations remain [2]. Gene expression profiling in breast cancer can be used for molecular classification based on an improved understanding of breast cancer heterogeneity [3]. The molecular subtypes of breast cancer have been shown to influence patient management. Various proteins have been proposed as potential prognostic biomarkers based on the analysis of the correlation between protein expression and proven clinicopathological prognostic factors [4].

Cortactin is a cytoskeleton-related protein, encoded by the CTTN gene (formerly known as EMS1), located at chromosome 11q13 [5]. Cortactin is also a phosphorylated protein originally identified as a substrate of the Src tyrosine kinase [6]. Cortactin plays a key role in regulating cell migration and invasion, processes that are closely associated with cancer progression [5,7,8]. Cortactin binds to the actin-associated protein (Arp) 2/3 complex and activates actin polymerization. Cortactin regulates invadopodia formation by binding and activating the Arp 2/3 complex, thereby promoting cell motility and tumor metastasis [9]. The Arp2/3 complex is involved in the regulation of the actin cytoskeleton, including endocytosis, vesicular transport, cellular invasion, and metastasis [10]. Cortactin plays a central role in the development and maturation of invadopodia and drives the degradation of the extracellular matrix in invasive cancer cells [7,11].

Overexpression of cortactin due to the amplification of the chromosome 11q13 locus has been observed in various cancers, such as head and neck squamous cell carcinoma [12], hepatocellular carcinoma [13], colon cancer [14], and bladder cancer [15]. In breast cancer, 11q13 amplification is found in up to 15% of the cases [16,17] and is associated with poor prognosis [18]. Although studies on the correlation between cortactin expression and prognosis in breast cancer have been published, any correlation between cortactin expression and prognosis according to the molecular subtype has not been evaluated. Therefore, we analyzed the correlation between the differential expression of cortactin and clinicopathological factors for each molecular subtype and evaluated its role as a biomarker that can predict prognosis.

## 2. Materials and Methods

### 2.1. Patient Selection and Clinical Data Collection

According to the Declaration of Helsinki principles, approval was obtained from the Institutional Review Board of Hanyang University Hospital before conducting the study (IRB No. HYUH 2021-12-014-001). The requirement for informed consent was waived because this study involved minimal risk to the participants and did not infringe upon their rights. Specimens were obtained from 541 patients with invasive breast carcinoma who underwent surgery at Hanyang University Hospital (Seoul, Republic of Korea) between February 2003 and January 2017. Patients with incomplete clinical follow-up or no available paraffin blocks were excluded, and a final cohort of 506 patients with breast cancer was established.

### 2.2. Tissue Microarray (TMA) Construction

A manual tissue microarray was used for TMA construction from archival formalin-fixed and paraffin-embedded tissue blocks. The most representative portion of the tumor was selected using a light microscope. Previously marked lesions from each donor blocks were punched out in a cylindrical form with a diameter of 3 mm and inserted into a pre-made paraffin recipient block comprising 6 × 5 samples in each TMA block.

### 2.3. Immunohistochemical (IHC) Staining

Immunohistochemistry (IHC) was performed on the sections using a Benchmark XT Automated Staining System (Ventana Medical Systems, Tucson, AZ, USA). Anti-cortactin (11381-1-AP, 1:400, Proteintech, Rosemont, IL, USA) was used as the primary antibody according to the manufacturer’s instructions. Heat-induced epitope retrieval was performed using CC1 Tris-EDTA buffer (Ventana Medical Systems, Tucson, AZ, USA). The signal was detected using OptiView DAB IHC Detection Kit (Ventana Medical Systems, Tucson, AZ, USA) containing an endogenous peroxidase blocker. Modified Mayer’s hematoxylin (hematoxylin II) was used as a counterstain. Staining, in which all procedures were performed except the primary antibody treatment, served as the negative control.

### 2.4. Grading for Cortactin Immunoreactivity

The staining pattern of normal breast tissue was heterogeneous, with no staining in basal cells and weak intensity staining in ductal cells. Representative microscopic images of cortactin expression in normal tissue are shown in Appendix A. Cortactin immunoreactivity was graded using an immunoreactivity score (IRS). Quantification of IRS for cortactin expression was performed by multiplying the intensity score by the percentage of immunoreactive cells. The specimens were visually evaluated by two pathologists, and intensity scores on a 3-point scale (1: weak, 2: moderate, 3: strong) were assigned. The percentage of immunoreactive cells was divided into 4 groups with scores ranging from 0 to 4 where scores 0, 1, 3, and 4 represented 0%, 1–25%, 51–75%, and 76–100%, respectively. Cortactin expression was classified into low cortactin or high cortactin groups by the IRS score with IRS ≤ 4 indicating low cortactin and IRS > 4 high cortactin expression. Representative microscopic images of cortactin expression are shown in Figure 1. There is no standard cut-off value for cortactin expression. Therefore, we performed receiver operating characteristic (ROC) curve analysis and determined the optimal cut off value as the point maximizing Youden’s index for each type of breast cancer.

### 2.5. Statistical Analysis

Chi-squared and Fischer’s exact tests were used to assess the potential association between cortactin expression and other clinicopathological parameters and categorical variables. The Mann–Whitney U test and Student’s t-test were used to analyze the association of cortactin expression and clinicopathological parameters with continuous variables. Overall survival (OS) was defined as the duration from surgical treatment to death, and disease-free survival (DFS) was defined as the duration from surgical treatment to first clinical or pathological recurrence. The Kaplan–Meier method with a log-rank test was used to construct survival curves, and univariate and multivariate Cox proportional hazard ratio models were used to determine the significant prognostic variables. Statistical significance was set at *p* < 0.05. Statistical analyses were performed using R (version 4.2.2).

## 3. Results

### 3.1. Clinicopathological Characteristics

The mean patient age was 52.9 years, and the mean tumor size was 2.6 cm. The numbers of patients with histologic Grades 1, 2, and 3 were 112, 226, and 168, respectively. Divided by molecular subtypes, 202, 109, 87, and 108 patients had the Luminal A, Luminal B, HER2-positive, and TNBC subtypes, respectively. According to the American Joint Committee on Cancer (AJCC) classification (8th edition), 178, 232, and 96 patients were Stage I, II, and III, respectively. The assessed clinicopathological characteristics included patient age, tumor size, histologic grade, histologic type, molecular type, pathologic T stage, pathologic N stage, AJCC stage, and patient survival. The clinicopathological characteristics of all patients and those with TNBC are summarized in Table 1 and Table 2.

### 3.2. Correlation between Cortactin Expression and Clinicopathological Parameters

Cytoplasmic cortactin expression was evaluated on TMA slides by two pathologists (Hwangkyu Son and Jae Kyung Myung). Of the 506 invasive breast carcinoma cases, 333 showed high cortactin expression, and 173 showed low cortactin expression following IHC staining. The correlations between cortactin expression and other clinicopathological parameters in all breast cancer specimens are summarized in Table 3. There was no correlation between cortactin expression, age, and tumor size. Histologic grade, molecular subtype, pT stage, pN stage, and AJCC stage also showed no correlation with cortactin expression. The patients were divided into non-TNBC and TNBC groups to evaluate the relevance of cortactin expression and related clinicopathological factors. However, cortactin-expression-related changes in clinicopathological factors were not observed in any group. Correlations between cortactin expression and other clinicopathological parameters in all patients and those with TNBC are summarized in Table 3 and Appendix A.

### 3.3. Correlation between Cortactin Expression and Patient Outcomes

Univariate Cox regression analysis of disease-free survival (DFS) in all patients with breast cancer showed that tumor size (*p* < 0.001), histologic grade (*p* = 0.017, *p* < 0.001, respectively), T stage (*p* < 0.001), N stage (*p* < 0.001), and AJCC stage (*p* = 0.004, *p* < 0.001, respectively) were significantly associated with DFS. In OS, tumor size (*p* < 0.001), histologic grade (Grade 3 vs. Grade 1; *p* = 0.001), molecular type (TNBC vs. Luminal A, Luminal B, HER2; *p* = 0.05), T stage (*p* < 0.001), N stage (*p* < 0.001), and AJCC stage (*p* = 0.031, *p* < 0.001, respectively) were significantly associated with OS. Multivariate Cox regression analysis of DFS in all patients with breast cancer showed that histologic grade (*p* = 0.043, *p* = 0.011, respectively) and N stage (*p* < 0.001) were significantly associated with DFS. Multivariate Cox regression analysis of OS in all patients with breast cancer showed that histologic grade (Grade 3 vs. 1; *p* = 0.009), T stage (*p* = 0.048) and N stage (*p* = 0.007) were significantly associated with OS. The results of this analysis are summarized in Table 4. Univariate Cox regression analysis for DFS and OS in patients with TNBC showed no statistically significant association with tumor size, histologic grade, molecular type, T stage, N stage, AJCC, or cortactin expression. Multivariate Cox regression analysis was performed on TNBC specimens. The T-stage and N-stage were significantly associated with DFS (*p* = 0.005, *p* = 0.043, respectively). The T-stage and cortactin expression were significantly associated with OS (*p* = 0.009, *p* = 0.006, respectively). The results of this analysis are summarized in Table 5.

The effect of cortactin expression on patient survival was also examined. OS and DFS were analyzed according to cortactin expression in all patients; however, the results were not statistically significant (*p* = 0.811, *p* = 0.394, respectively). In the TNBC and non-TNBC groups, the difference in survival rates according to cortactin expression was analyzed. In non-TNBC patients, the high-cortactin group had a poor prognosis, but no statistical significance could be drawn (*p* = 0.922, *p* = 0.181, respectively). Contrary to the results of the Kaplan–Meier curve analysis using all patients and non-TNBC patients, higher cortactin expression tended to be associated with better DFS and OS in patients with TNBC. However, the relation between cortactin expression and DFS was not statistically significant (*p* = 0.097), whereas a statistically significant association was observed between cortactin expression and OS (*p* = 0.007). The Kaplan–Meier curves for all patients with breast cancer, TNBC, and non-TNBC are summarized in Figure 2.

## 4. Discussion

The cortactin locus, encoded by the cortactin gene within chromosome region 11q13, is amplified in several human cancers [5]. Amplification of cortactin is found in 15% of primary breast cancers and 29% of head and neck squamous cell carcinomas. According to the literature, overexpression of cortactin is related to tumor invasion and metastasis, and has been reported to be associated with poor prognosis [19]. Cai et al., reported that cortactin overexpression was correlated with tumor invasion, and that cortactin expression was an independent prognostic factor for both OS and DFS in Stage II–III colorectal cancer [20]. In addition to clinical conclusions such as the predictive association of cortactin overexpression with prognosis, a previous study has shown that overexpression of cortactin causes tumor cell proliferation in vitro [21].

In breast cancer, in vivo and in vitro experimental data have been used to investigate the effects of CTTN expression on tumor development, growth, and metastasis. Van Rossum et al., generated mouse mammary tumor virus (MMTV)-cortactin transgenic mice and MMTV-cortactin/MMTV-cyclin D1 bi-transgenic mice to study the role of cortactin in mammary gland tumorigenesis [22]. Since the amplification of chromosome 11q13 was correlated with lymph node metastasis and increased mortality in patients with breast cancer, they studied the role of two genes located within this amplicon, CCND1 and CTTN, in the development of breast tumors using transgenic mice. They did not observe any synergistic effect of cortactin on cyclin-D1-induced mammary hyperplasia or carcinoma nor the development of distant metastasis. Li et al., reported that cortactin facilitates tumor metastasis in breast cancer by enhancing the interaction of tumor cells with endothelial cells and the invasion of tumor cells into bone tissues [23]. Bowden et al., reported that cortactin participates in tumor cell invasion in breast cancer via the invadopodia complex [24].

In addition to these experimental data, studies revealing a correlation between cortactin expression and prognosis in patients with breast cancer have also been published. Although previous studies have examined the correlation between cortactin expression and prognosis in patients with breast cancer, the prognostic significance of cortactin expression in breast cancer remains unclear. Dedes et al., studied the effects of cortactin expression and amplification in breast cancer using immunohistochemical staining and chromogenic in situ hybridization [25]. Neither cortactin expression nor gene amplification was associated with disease-free, metastasis-free, or overall survival. Sheen-Chen et al., reported the prognostic value of cortactin expression based on a tissue microarray of 99 patients with breast cancer [26]. In this study, cortactin expression did not show any significant relationship with the overall survival rate. The absence of any association between cortactin expression and the prognosis of patients with breast cancer in these two studies is probably explained by the fact that the molecular subtypes of breast cancer were not separately analyzed in these studies. This assumption is supported by a recent study where cortactin overexpression in a HER2-type breast cancer was found to be associated with poor clinical outcome like other carcinomas [27]. Unlike the HER2-type breast cancer, cortactin protein expression in TNBC breast cancer is associated with good prognosis based on our data. Based on these results, overexpression of cortactin in TNBC and HER2-type breast cancers is presumed to play different roles, or it is judged that different molecular mechanisms are involved. This observation needs to be supported by experimental studies in the future.

In this study, we evaluated cortactin expression in 506 cases of invasive breast carcinoma and investigated the correlation between cortactin expression, clinicopathological characteristics, and patient survival. In addition to evaluating patients with breast cancer, the same investigations were performed according to the difference in molecular subtype. There was no significant correlation between cortactin expression and other clinicopathological characteristics in patients with breast cancer. In survival analysis, patients with high cortactin expression showed poorer OS and DFS than those with low cortactin expression in all patients with breast cancer and non-TNBC patients. These results are reversed in patients with TNBC. Based on these results, we confirmed that the effect of cortactin expression on patient prognosis in the TNBC and non-TNBC patient groups was the opposite. In the TNBC group, increased expression correlated with better DFS and OS, and the statistical significance of OS was confirmed. Therefore, we hypothesized that cortactin expression might have an important effect on the prognosis of patients with TNBC.

Triple-negative breast cancer (TNBC) is a molecular subtype of breast cancer with negative expression of estrogen receptor (ER), progesterone receptor (PR), and human epidermal growth factor receptor-2 (HER2) [28]. TNBC was initially perceived as a clinical entity closely related to the basal-like subtype discovered through seminal gene expression microarray studies in the early 2000s [29]. TNBC is well known for its aggressive behavior and is characterized by onset at a younger age, high mean tumor size, higher-grade tumors, and an elevated rate of node metastasis [30]. In addition, TNBC is known for an early peak of recurrence between the first and third years after diagnosis, and more aggressive metastases that are more likely to occur in the viscera, particularly in the lungs and brain, and are less likely to spread to the bone [31]. In addition, TNBC is a highly invasive cancer with a high incidence of distant metastasis (approximately 46% of all patients) [32]. Although chemotherapy is the main systemic treatment for TNBC due to its unique molecular phenotype, the efficacy of conventional postoperative adjuvant chemoradiotherapy is poor [33]. To predict the treatment and prognosis of patients with TNBC, it is necessary to identify new targets that differ from those of non-TNBC patients. Therefore, the effect of cortactin expression on the prognosis of patients with TNBC is clearly significant. One limitation of this study is that it was a retrospective study performed with cases collected from a single center. Additionally, experimental studies on the precise molecular mechanisms underlying cortactin expression in patients with TNBC are also required.

In conclusion, we investigated the clinicopathological significance of cortactin expression in 506 cases of invasive breast carcinoma. Strong cortactin expression was significantly associated with better OS in patients with TNBC, unlike in non-TNBC patients.

## Figures and Tables

**Figure 1 diagnostics-13-02876-f001:**
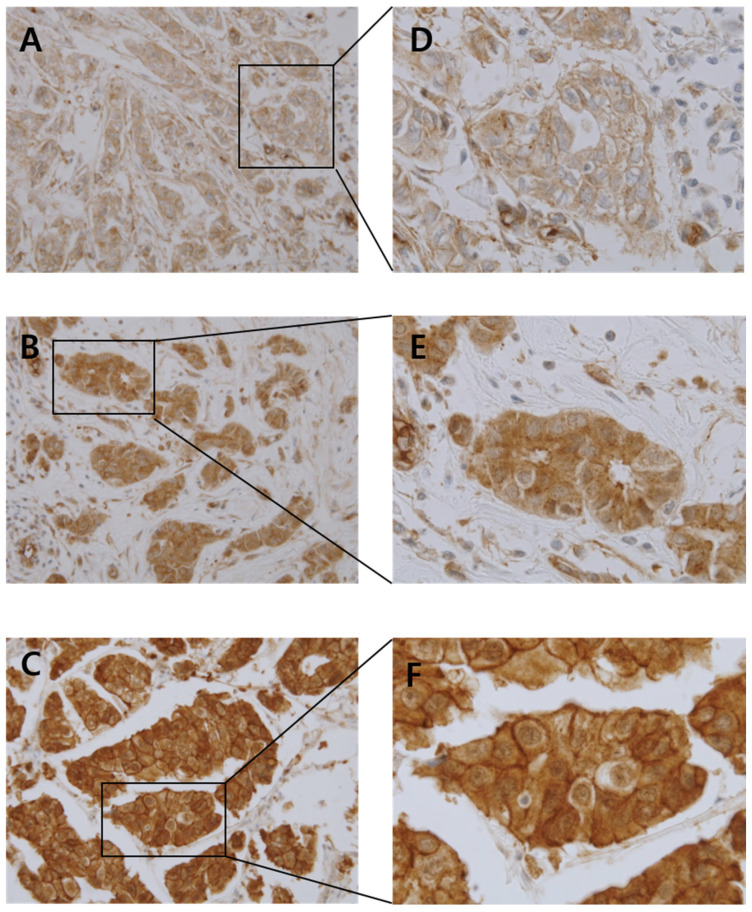
Representative photomicrographs of intensity score with cortactin staining on tissue microarrays ((**A**): weak expression, (**B**): moderate expression, (**C**): strong expression, original magnification ×400 (**A**–**C**), original magnification ×1000 (**D**–**F**)).

**Figure 2 diagnostics-13-02876-f002:**
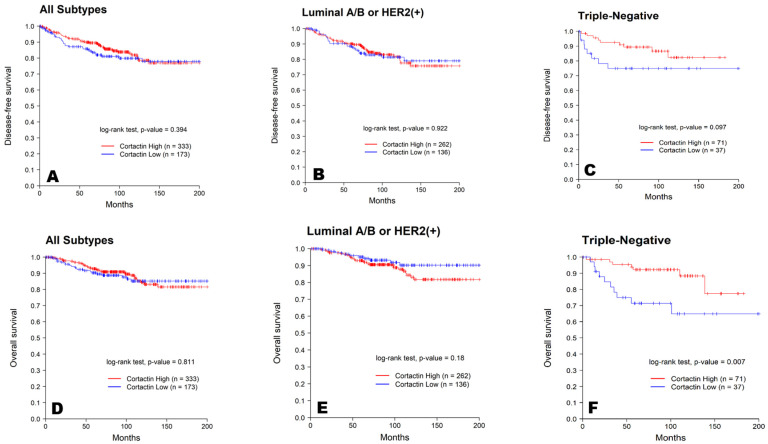
Kaplan–Meier survival curves for DFS (**A**–**C**) and OS (**D**–**F**) of all, non-TNBC, and TNBC patients based on cortactin expression. The blue line represents low expression of cortactin, and the red line represents high expression of cortactin.

**Table 1 diagnostics-13-02876-t001:** Baseline characteristics of all enrolled patients (*n* = 506).

Clinicopathological Characteristics		Value (%)
Age (years, mean ± SD), (range)		52.6 ± 11.1, (27–83)
Size (cm, mean ± SD), (range)		2.6 ± 1.8, (0.12–16)
Sex		
	Male	0 (0%)
	Female	506 (100%)
Histologic grade		
	Grade 1	112 (22.1%)
	Grade 2	226 (44.7%)
	Grade 3	168 (33.2%)
Histologic type		
	Invasive breast carcinoma, No special type	474 (93.7%)
	Lobular carcinoma	22 (4.3%)
	Other	10 (2.0%)
Molecular type		
	Luminal A	202 (40.0%)
	Luminal B	109 (21.5%)
	HER2	87 (17.2%)
	Triple negative	108 (21.3%)
	Unknown	0 (0.0%)
pT stage		
	T1	229 (45.3%)
	T2	233 (46.0%)
	T3	30 (5.9%)
	T4	14 (2.8%)
pN stage		
	N0	319 (63.0%)
	N1	108 (21.4%)
	N2	44 (8.7%)
	N3	35 (6.9%)
AJCC stage (8th)		
	I	178 (35.2%)
	II	232 (45.8%)
	III	96 (19.0%)
Recurrence		
	Recurrence	93 (18.4%)
	No recurrence	413 (81.6%)
Death		
	Death	57 (11.2%)
	Alive	449 (88.7%)

Abbreviations: SD, standard deviation; AJCC, American Joint Committee on Cancer.

**Table 2 diagnostics-13-02876-t002:** Baseline characteristics of triple-negative subtype patients (*n* = 108).

Clinocpathological Characteristics		Value (%)
Age (years, mean ± SD), (range)		50.9 ± 10.6 (27–75)
Size (cm, mean ± SD), (range)		3.1 ± 2.3 (0.7–16)
Sex		
	Male	0 (0%)
	Female	108 (100%)
Histologic grade		
	Grade 1	9 (8.3%)
	Grade 2	23 (21.3%)
	Grade 3	76 (70.4%)
Histologic type		
	Invasive breast carcinoma, No special type	101 (93.5%)
	Lobular carcinoma	2 (1.9%)
	Other	5 (4.6%)
pT stage		
	T1	30 (27.8%)
	T2	65 (60.2%)
	T3	8 (7.4%)
	T4	5 (4.6%)
pN stage		
	N0	69 (63.9%)
	N1	22 (20.4%)
	N2	7 (6.5%)
	N3	10 (9.2%)
AJCC stage (8th)		
	I	24 (22.2%)
	II	62 (57.4%)
	III	22 (20.4%)
Recurrence		
	Recurrence	19 (17.6%)
	No recurrence	89 (82.4%)
Death		
	Death	17 (15.7%)
	Alive	91 (84.3%)

Abbreviations: SD, standard deviation; AJCC, American Joint Committee on Cancer.

**Table 3 diagnostics-13-02876-t003:** Correlations between cortactin expression and clinicopathological factors in breast cancer (*n* = 506).

Parameter		Cortactin Expression		*p* Value
		High Group (*n* = 333) No. (%)	Low Group (*n* = 173) No. (%)	
Age				0.597 †
(years, mean ± SD), (range)		53.1 ± 11.3 (28–83)	52.4 ± 10.9 (27–78)	
Tumor size				0.775 †
(cm, mean ± SD), (range)		2.6 ± 1.9 (0.12–16)	2.6 ± 1.6 (0.5–11)	
Histologic grade				0.238
	Grade 1	70 (21.0%)	42 (24.3%)	
	Grade 2	144 (43.3%)	82 (47.4%)	
	Grade 3	119 (35.7%)	49 (28.3%)	
Molecular type				0.312
	Luminal A	125 (37.6%)	77 (44.5%)	
	Luminal B	79 (23.7%)	30 (17.3%)	
	HER2	58 (17.4%)	29 (16.8%)	
	Triple negative	71 (21.3%)	37 (21.4%)	
pT stage				0.615
	T1	151 (45.4%)	78 (45.1%)	
	T2	150 (45.0%)	83 (48.0%)	
	T3	23 (6.9%)	7 (4.0%)	
	T4	9 (2.7%)	5 (2.9%)	
pN stage				0.761
	N0	209 (62.8%)	110 (63.6%)	
	N1	75 (22.5%)	33 (19.1%)	
	N2	27 (8.1%)	17 (9.8%)	
	N3	22 (6.6%)	13 (7.5%)	
AJCC stage (8th)				0.623
	I	122 (36.6%)	56 (32.4%)	
	II	150 (45.1%)	82 (47.4%)	
	III	61 (18.3%)	35 (20.2%)	

Abbreviations: †, Mann–Whitney U test.

**Table 4 diagnostics-13-02876-t004:** Cox regression analysis for disease-free survival and overall survival in all patients with breast carcinoma (*n* = 506).

Variables	Disease-Free Survival			Overall Survival		
	HR	95% CI	*p* Value	HR	95% CI	*p* Value
Univariate analysis						
Age (per 1 year)	0.990	(0.968–1.011)	0.307	1	(0.976–1.025)	0.998
Size (per 1 cm)	1.207	(1.111–1.311)	<0.001	1.255	(1.144–1.376)	<0.001
Histologic grade 2	2.670	(1.190–5.989)	0.017	2.766	(0.957–8.000)	0.060
(vs. 1)						
Histologic grade 3	3.923	(1.746–8.816)	0.001	5.624	(1.981–15.966)	0.001
(vs. 1)						
TNBC subtype	1.095	(0.641–1.871)	0.741	1.764	(0.998–3.112)	0.051
(vs. Luminal A, Luminal B, HER2)						
pT stage 4	5.058	(2.042–12.53)	<0.001	6.779	(2.695–17.05)	<0.001
(vs. 1, 2, 3)						
pN stage 1, 2, 3	3.114	(1.979–4.897)	<0.001	2.802	(1.643–4.777)	<0.001
(vs. 0)						
AJCC stage II (vs. I)	2.581	(1.351–4.929)	0.004	2.401	(1.083–5.323)	0.031
AJCC stage III (vs. I)	5.210	(2.658–10.214)	<0.001	5.967	(2.679–13.287)	<0.001
Cortactin expression low (vs. high)	1.218	(0.774–1.915)	0.394	1.069	(0.620–1.842)	0.811
Multivariate						
analysis						
Age (per 1 year)	0.998	0.978–1.019	0.874	1.010	0.986–1.034	0.426
Size (per 1 cm)	1.076	0.969–1.195	0.161	1.119	0.997–1.257	0.057
Histologic grade 2 (vs. 1)	2.320	1.028–5.235	0.045	2.455	0.843–7.153	0.103
Histologic grade 3 (vs. 1)	2.953	1.286–6.778	0.011	4.152	1.434–12.026	0.009
pT stage 4 (vs. 1, 2, 3)	2.325	0.835–6.471	0.108	2.905	1.008–8.370	0.048
pN stage 1, 2, 3 (vs. 0)	2.595	1.624–4.146	<0.001	2.139	1.230–3.718	0.007
Cortactin expression low (vs. high)	1.360	(0.861–2.149)	0.187	1.221	0.705–2.117	0.476

**Table 5 diagnostics-13-02876-t005:** Cox regression analysis of disease-free survival and overall survival in triple-negative subtype of breast cancer only (*n* = 108).

Variables	Disease-Free Survival			Overall Survival		
	HR	95% CI	*p* Value	HR	95% CI	*p* Value
Univariate analysis						
Age (per 1 year)	1.005	(0.959–1.052)	0.843	0.997	(0.951–1.045)	0.903
Size (per 1 cm)	1.005	(0.810–1.247)	0.965	0.997	(0.791–1.257)	0.981
Histologic grade 2 (vs. 1)	2.232	(0.261–19.11)	0.464	1.752	(0.195–15.73)	0.617
Histologic grade 3 (vs. 1)	1.477	(0.191–11.44)	0.709	1.666	(0.216–12.84)	0.624
pT stage 4 (vs. 1,2,3)	3.921	(0.889–17.29)	0.071	4.251	(0.955–18.92)	0.058
pN stage 1, 2, 3 (vs. 0)	2.323	(0.8946–6.034)	0.083	1.502	(0.571–3.952)	0.41
AJCC stage II (vs. I)	1.720	(0.377–7.853)	0.484	0.6478	(0.195–2.154)	0.479
AJCC stage III (vs. I)	2.575	(0.499–13.293)	0.259	1.2367	(0.332–4.611)	0.752
Cortactin expression	2.199	(0.847–5.71)	0.105	3.511	(1.331–9.258)	0.011
low (vs. high)						
Multivariate analysis						
Age (per 1 year)	1.003	0.958–1.050	0.891	0.997	0.951–1.045	0.910
Size (per 1 cm)	0.821	0.619–1.087	0.169	0.827	0.606–1.127	0.228
Histologic grade 2 (vs. 1)	2.001	0.219–18.263	0.539	2.078	0.214–20.175	0.528
Histologic grade 3 (vs. 1)	1.100	0.129–9.354	0.933	1.816	0.214–15.403	0.584
pT stage 4 (vs. 1, 2, 3)	16.385	2.322–115.604	0.005	13.419	1.917–93.936	0.009
pN stage 1, 2, 3 (vs. 0)	2.926	(1.036–8.262)	0.043	1.556	0.560–4.320	0.396
Cortactin expression	2.529	(0.946–6.763)	0.065	4.058	1.489–11.056	0.006
low (vs. high)						

Abbreviations: AJCC, American Joint Committee on Cancer; HR, hazard ratio; CI, confidence interval.

## Data Availability

Not applicable.

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
