# Peer review of "Effects of Cortactin Expression on Prognosis in Patients with Breast Cancer"

_diagnostics, 2023, doi:10.3390/diagnostics13182876_

Round 1

Reviewer 1 Report

The manuscript "The effects of cortactin expression on prognosis in patients with breast cancer" by Son et al., is an interesting study focused on the expression of cortactin in breast cancer samples.  This manuscript highlighted the effects of cortactin in breast cancer but lacks the merit of publication. 

1- The language of this manuscript needs changes by an expert. 

2- In abstract, 490 patients are mentioned which is different from 508. 

3- This study does not have data of control tissue so lacks the scientific significance.  

The language of this manuscript needs changes by an expert. 

Author Response

1- The language of this manuscript needs changes by an expert.

Answer:

Although, English proofreading was done through editage and the draft was submitted. As you pointed out, English proofreading was carried out once more. Therefore, after two English corrections through editage, many expressions were changed.

2- In abstract, 490 patients are mentioned which is different from 508.

Answer:   

As pointed out by reviewers, the total number of patients was changed to 506, excluding male patients. The total number of patients was 506, which was corrected in the abstract and text. As stated in the text, the initial registered patients were 541, but the final registered patients were 506 because patients whose clinical progress could not be confirmed and patients who could not obtain a paraffin block were excluded. All patients were changed to 506 in the entire body including Table  except for this part.

3- This study does not have data of control tissue so lacks the scientific significance.

Answer:   

As pointed out, negative control was described as follows. "Staining in which all procedures were performed except the the primary antibody treatment served as the negative control"

The positive control group was set up through the following method but was omitted because these expressions are not generally marked. If these expressions need to be included, I will revise them again. Positive control: Human breast cancer tissue was used as a positive control by referring to the manufacturer's data sheet, and conditions such as appropriate primary antibody concentration and pretreatment were set using this.

Reviewer 2 Report

The present study aims to investigate the role of cortactin expression in breast cancer prognosis. The authors have found that in patients with TNBC, the high-cortactin group had a better prognosis in disease-free survival and overall survival, therefore they concluded that cortactin expression may be a good biomarker for predicting the prognosis of patients with TNBC.

The work has an interest, as it provides some evidence on a highly aggressive breast cancer subtype, for which no targeted therapy exists. The manuscript is well-written, and the results are well-described and presented. However, as it is noted by the authors, the study performed with cases collected from a single center and the observations may be biased. In addition, there are some points requiring correction and/or clarification, as follows.

1.       According to literature, cortactin plays a central role in the development and maturation of invadopodia and drives the degradation of the extracellular matrix in invasive cancer cells, therefore it is surprising the finding that in patients with TNBC, the high-cortactin group had a better prognosis in disease-free survival and overall survival. The authors should discuss this.

2.       Clearly describe the number of patients included in the study (see lines 13, 18, 59, and elsewhere in text and tables).

3.       Since the study suggested a role of cortactin in TNBC, it is better to incorporate supplementary table 1 in main text.

4.       Line 42: The sentence “Cortactin contains an N-terminal acidic region that cortactin binds to and activates the Arp2/3 complex” should be revised and corrected.

5.       Add the age range of patients in all tables.

6.       It is better to remove male patients from all statistics. Of course, no male patients with TNBC existed and this permits to the authors to conclude for the role of cortactin as TNBC biomarker.

Author Response

  1. According to literature, cortactin plays a central role in the development and maturation of invadopodia and drives the degradation of the extracellular matrix in invasive cancer cells, therefore it is surprising the finding that in patients with TNBC, the high-cortactin group had a better prognosis in disease-free survival and overall survival. The authors should discuss this.

Answer:          

It is known that the expression of cortactin plays an important role in the expression and maturation of invadopodia by dissolving the extracellular matrix in some carcinomas. In breast cancer, the role of cortactin related to this invasive growth has been cited in one literature recently. The relationship between cortactin expression and the invasive growth of breast cancer cells has been reported in HER2-positive breast cancer, but not in TNBC. Therefore cortactin expression is judged to have a different effect on prognosis according to different molecular types. Through which molecular mechanism, it is judged that the role of cortaction protein expression in TNBC as a good prognostic factor and the role different from breast cancer of other molecular types should be revealed through further experiments. With this content, the following phrases were described in the discussion section.

"The absence of any assocation between cortactin expression and the prognosis of patients with breast cancer in these two studies is probably because the molecular subtypes of breast cancer were not separately analyzed in these studies. This assumption is supported by a recent study where cortactin overexpression in HER2 type breast cancer was found to be associated with poor clinical outcome like other carcinomas. Unlike HER2 type breast cancer, cortactin protein expression in TNBC breast cancer is associated with good prognosis based on our data. Based on these results, overexpression of cortatin in TNBC and HER2 type breast cancer is presumed to play different roles, or it is judged that different molecular mechanisms are involved. This observation needs to be supported by experimental studies in the future."

  1. Clearly describe the number of patients included in the study (see lines 13, 18, 59, and elsewhere in text and tables).

Answer:          

As pointed out by reviewers, the total number of patients was changed to 506, excluding male patients. The total number of patients was 506, which was corrected in the abstract and text. As stated in the text, the initial registered patients were 541, but the final registered patients were 506 because patients whose clinical progress could not be confirmed and patients who could not obtain a paraffin block were excluded. All patients were changed to 506 in the entire body including Table  except for this part.

  1. Since the study suggested a role of cortactin in TNBC, it is better to incorporate supplementary table 1 in main text.

Answer:          

As pointed out, supplementary table 1 has been moved to the main text as a Table 2.

  1. Line 42: The sentence “Cortactin contains an N-terminal acidic region that cortactin binds to and activates the Arp2/3 complex” should be revised and corrected.

Answer:

As pointed out, modified as follows.

"Cortactin binds to the actin-associated protein (Arp) 2/3 complex and activates actin polymerization. Cortactin regulates invadopodia formation by binding and activating the Arp 2/3 complex, thereby promoting cell motility and tumor metastasis."

  1. Add the age range of patients in all tables.

Answer:

As pointed out, the distribution of age and size are indicated in all tables.

  1. It is better to remove male patients from all statistics. Of course, no male patients with TNBC existed and this permits to the authors to conclude for the role of cortactin as TNBC biomarker.

Answer:

:As pointed out, two male patients were excluded from the total patient group. Statistical processing was repeated except for two male patients, and the number of patients included in the table and text was changed. The total number of patients was changed from 508 to 506, excluding male patients. In the TNBC patient group, there was no change in the figures because male patients were not included.

Round 2

Reviewer 1 Report

Thank you for addressing the suggestions in Round 1. Coming back to my third point " This study does not have data of control tissue so lacks the scientific significance". Don't you have data of normal healthy cells, next to the cancerous cells of same tissues? It would support your results. 

Author Response

Thank you for the detailed review and comments.

The attached photo is a photo evaluating cortactin expression in normal breast tissue observed around tumor cells. The staining pattern was heterogeneous, with no staining in basal cells and weak intensity staining in ductal cells. These staining patterns and explanations inserted into the text (Grading for cortactin immunoreactivity) and representative microscopic images of cortactin expression in normal tissue are shown in supplementary Figure 1.

Round 3

Reviewer 1 Report

Thank you for the addressing comments/suggestions.